# Treatment Sequences in Patients with Recurrent or Metastatic Head and Neck Squamous Cell Carcinoma: Cetuximab Followed by Immunotherapy or Vice Versa

**DOI:** 10.3390/cancers14102351

**Published:** 2022-05-10

**Authors:** Chuan-Chien Yang, Ching-Feng Lien, Tzer-Zen Hwang, Chih-Chun Wang, Chien-Chung Wang, Yu-Chen Shih, Shyh-An Yeh, Meng-Che Hsieh

**Affiliations:** 1Department of Otolaryngology, E-Da Hospital, Kaohsiung 82445, Taiwan; ed101194@edah.org.tw (C.-C.Y.); ed106789@edah.org.tw (C.-F.L.); ed103548@edah.org.tw (T.-Z.H.); ed103747@edah.org.tw (C.-C.W.); philips115@isu.edu.tw (C.-C.W.); 2College of Medicine, I-Shou University, Kaohsiung 82445, Taiwan; 107033@kh.dhp.edu.tw (Y.-C.S.); 109046@kh.dhp.edu.tw (S.-A.Y.); 3Department of Otolaryngology, E-Da Cancer Hospital, Kaohsiung 82445, Taiwan; 4Department of Radiation Oncology, E-Da Hospital, Kaohsiung 82445, Taiwan; 5Department of Hematology-Oncology, E-Da Cancer Hospital, Kaohsiung 82445, Taiwan

**Keywords:** cetuximab, immunotherapy, recurrent or metastatic head and neck squamous cell carcinoma, treatment sequence, survival, prognosis

## Abstract

**Simple Summary:**

As the treatment advances, there were several novel agents developed for R/M HNSCC, including an anti-epidermal growth factor receptor antibody, cetuximab, and an anti-programmed death-1 immune checkpoint inhibitor, pembrolizumab and nivolumab. To date, little was known regarding the optimal treatment sequences. Our observational study demonstrated that median overall survival was 23.7 months versus 22.8 months in Cet-IO and IO-Cet, respectively (*p* = 0.484). The overall response rate (ORR) were 73% in Cet-IO versus 37% in IO-Cet (*p* = 0.002). Both Cet-IO and IO-Cet are effective in R/M HNSCC patients with insignificant survival differences. The higher response rate of Cet-IO might render it to be considered in patients with large tumor burdens and urgent needs for treatment responses. Our conclusion can be real-world evidence for clinical decision-making.

**Abstract:**

Background: The prognosis was poor when patients had recurrent or metastatic head and neck squamous cell carcinoma (R/M HNSCC). Herein, we conducted an observational study of cetuximab followed by immunotherapy (Cet-IO) versus immunotherapy followed by cetuximab (IO-Cet) in patients with R/M HNSCC. Methods: Patients who were diagnosed with R/M HNSCC and treated with a sequential cetuximab-containing regimen and immunotherapy-containing regimen were enrolled in our study. Kaplan-Meier curves were estimated for progression-free survival (PFS) and overall survival (OS). Results: A total of 75 patients were enrolled in our study for oncologic outcomes evaluation, with 40 patients in Cet-IO and 35 patients in IO-Cet. The median PFS1 was 5.1 months in Cet-IO and 4.5 months in IO-Cet (*p* = 0.777) and the median PFS2 was 16.5 months in Cet-IO and 11.4 months in IO-Cet (*p* = 0.566). The median OS was 23.7 months versus 22.8 months in Cet-IO and IO-Cet, respectively (*p* = 0.484). The overall response rate (ORR) were 73% in Cet-IO versus 37% in IO-Cet (*p* = 0.002). Multivariate analysis demonstrated that the treatment sequences, Cet-IO or IO-Cet, were insignificantly different with survival. Conclusion: Both Cet-IO and IO-Cet are effective in R/M HNSCC patients with insignificant survival differences. The higher ORR of Cet-IO might render it to be considered in patients with large tumor burdens and urgent needs for treatment responses. Further prospective studies are merited to validate our conclusions.

## 1. Introduction

Head and neck squamous cell carcinoma (HNSCC) is the sixth most common cancer, arising from the oral cavity, oropharynx, hypopharynx, and larynx [1]. The Global Burden of Disease study estimated 890,000 patients with a new diagnosis of HNSCC in 2017, representing 5.3% of all cancer cases [2]. The prognosis was poor when patients had recurrent or metastatic HNSCC (R/M HNSCC). The median overall survival (OS) of R/M HNSCC was around 8–10 months before 2005 [3]. As the treatment advances, there were several novel agents developed for R/M HNSCC [4]. Cetuximab, an anti-epidermal growth factor receptor antibody, was the first targeted therapy for R/M HNSCC. The EXTREME study demonstrated that cisplatin, 5-fluorouracil (5-FU) plus cetuximab followed by cetuximab maintenance weekly had significantly longer survival than cisplatin plus 5-FU [5]. The overall response rate (ORR) increased from 20% to 36%, median progression-free survival (PFS) prolonged from 3.3 to 5.6 months, and median OS extended from 7.4 to 10.1 months. Another novel agent for R/M HNSCC is an anti-programmed death-1 immune checkpoint inhibitor, pembrolizumab, which is introduced as a first-line treatment for R/M HNSCC. Keynote-048 found pembrolizumab plus platinum and 5-fluorouracil was effective for all R/M HNSCC patients and pembrolizumab monotherapy was effective for programmed death-ligand 1 (PD-L1)-positive R/M HNSCC patients [6]. Pembrolizumab alone significantly improved OS versus cetuximab with chemotherapy in the combined positive score (CPS) of 1 or more population (12.3 months vs. 10.3 months). Pembrolizumab with chemotherapy also significantly improved OS versus cetuximab with chemotherapy in the total population (13.0 months vs. 10.7 months). Thus, based on these conclusions, current guidelines all suggested that cetuximab with chemotherapy and pembrolizumab with/without chemotherapy were both standard first-line treatments for R/M HNSCC [7].

As for subsequent therapy, little was known regarding the treatment sequences. For patients with first-line cetuximab-containing treatment, immunotherapy might be a useful second-line regimen [8]. Checkmate 141 demonstrated that nivolumab as a second-line treatment resulted in significantly longer OS in patients with platinum-refractory R/M HNSCC [9]. Among these patients, 60% were treated with the previous first-line cetuximab-containing regimen. The median OS was 7.5 months in the nivolumab group versus 5.1 months in the group that received standard therapy. Keynote-040 also confirmed the significant prolongation of OS of pembrolizumab in patients with platinum-refractory R/M HNSCC [10]. Among these patients, 57.5% were treated with the previous first-line cetuximab-containing regimen. Median OS was 8.4 months with pembrolizumab and 6.9 months with the standard of care. On contrary, for patients with the first-line pembrolizumab-containing regimen, there was no clear evidence focusing on the later-line treatment. Subgroup analysis of Keynote-048 showed that less than 20% of patients receiving pembrolizumab or pembrolizumab plus chemotherapy were treated with a cetuximab-containing regimen. Thus, no conclusive results were suggested. Given that optimal treatment sequences were undetermined, we conducted an observational study of cetuximab followed by immunotherapy (Cet-IO) versus immunotherapy followed by cetuximab (IO-Cet) in patients with R/M HNSCC.

## 2. Materials and Methods

### 2.1. Patients

Patients who were diagnosed with R/M HNSCC from 2017 to 2020 at E-Da Hospital were reviewed. R/M HNSCC patients who were treated with a sequential cetuximab-containing regimen and immunotherapy-containing regimen or vice versa were enrolled in our study. Patients were stratified by treatment sequences, labeled as Cet-IO and IO-Cet groups. Cet-IO group referred to patients was treated with first-line cetuximab plus cisplatin and 5-fluorouracil (5-FU), followed by second-line immunotherapy with/without taxane. IO-Cet group referred to patients was treated with first-line pembrolizumab plus cisplatin and 5-FU, followed by second-line cetuximab plus a taxane. Treatment sequences were decided at the physicians’ discretion. Second-line immunotherapy was limited to pembrolizumab or nivolumab, which was determined by PD-L1 expression levels of primary tumors. Most tissues were resected specimens and the others were biopsies from patients with de novo metastatic HNSCC. PD-L1 scoring was calculated as the percentage of tumor cells exhibiting positive membrane staining at any intensity. Tumor proportional score (TPS) and combined positive score (CPS) PD-L1 expression was determined by Dako 22C3 pharmDx immunohistochemistry (IHC) assay, while tumor cell (TC) PD-L1 expression was determined by Dako 28–8 pharmDx IHC assay. High PD-L1 expression was defined as either TPS ≥ 50%, CPS ≥ 1, or TC ≥ 10%. Low PD-L1 expression was defined as TPS < 50%, CPS < 1 or TC < 10%. If TPS PD-L1 ≥ 50% or CPS PD-L1 ≥ 1, pembrolizumab might be suggested. If tumor cell (TC) PD-L1 ≥ 10%, nivolumab might be used in our patients. Taxane was confined into paclitaxel or docetaxel, which was also decided at physicians’ discretion. The patients’ clinical and laboratory data were collected from medical records. Inclusion criteria were pathologically confirmed R/M HNSCC and received both cetuximab and immunotherapy sequentially. Exclusion criteria were previous history of cetuximab or immunotherapy before R/M HNSCC, cetuximab or immunotherapy as third- or later-line treatment, rapid progression within six months after curative platinum-based concurrent chemoradiotherapy, and irregular follow-up intervals. This was a retrospective observational study, which was exempt from requiring consent. This study was approved by the E-Da Hospital Institutional Review Board (EMPR-110-167) and was conducted in accordance with the Declaration of Helsinki.

### 2.2. Treatment Protocols

For first-line cetuximab plus cisplatin and 5-FU, patients were treated with a 4-week cycle of cisplatin 70–100 mg/m^2^ on day 1 of each cycle and 5-FU 700–1000 mg/m^2^ on day 1–4 of each cycle plus cetuximab 400 mg/m^2^ loading at day 1 of cycle 1 and then 250 mg/m^2^ weekly on subsequent administration. For first-line pembrolizumab plus cisplatin and 5-FU, patients were treated with a 3-week cycle of cisplatin 70–100 mg/m^2^ on day 1 of each cycle and 5-FU 700–1000 mg/m^2^ on day 1–4 of each cycle plus pembrolizumab 2 mg/kg. For second-line cetuximab plus taxane, patients were treated with a 3-week cycle of paclitaxel 175 mg/m^2^ or docetaxel 75 mg/m^2^ plus cetuximab 400 mg/m^2^ loading then 250 mg/m^2^ weekly. For second-line immunotherapy with/without taxane, patients were treated with a 3-week cycle of pembrolizumab 2 mg/kg or a 2-week cycle of nivolumab 3 mg/kg with/without a 3-week cycle of paclitaxel 175 mg/m^2^ or docetaxel 75 mg/m^2^. Dose modification could be adjusted according to patients’ comorbidities and treatment adverse effects. Carboplatin was allowed in substitution for cisplatin if unfit renal function. Computed tomography was arranged for evaluation of the treatment response every 3–4 months. Treatment was continued in responding or stable patients until disease progression or unacceptable toxicity.

### 2.3. Statistical Analysis

All the clinical and basic characteristics were collected from a medical chart review and presented with frequencies. Chi-square tests were performed to compare the differences between Cet-IO and IO-Cet. Statistical analyses were calculated by using SPSS. The oncologic outcomes were presented with progression-free survival 1 (PFS1), PFS2, overall survival (OS), overall response rate (ORR), and disease control rate (DCR). PFS1 was measured from the first day of chemotherapy administration until the date of tumor progression after first-line treatment or final follow-up, while PFS2 was measured from the first day of chemotherapy administration until the date of tumor progression after second-line treatment or final follow-up. OS was calculated as the time from the first day of chemotherapy administration until the date of death from any cause or final follow-up. Objective response criteria were evaluated according to the RECIST 1.1 guidelines, including complete response (CR), partial response (PR), stable disease (SD), and progressive disease (PD). ORR was defined as CR plus PR, and DCR was defined as CR, PR, plus SD. Kaplan–Meier curves were depicted for survival. Multivariate analysis with Cox regression was also conducted using “enter” selection to adjust for the effects of potential confounders. All *p* values were two-sided and considered to have significance if *p* values < 0.05.

## 3. Results

### 3.1. Patients Characteristics

A total of 75 patients were enrolled in our study for oncologic outcomes evaluation. The median age of our patients is 54 years and the median follow-up period was 16 months. Baseline characteristics were presented in Table 1. In general, most patients were male (93%) in gender with 65% of patients being younger than 60 years. The majority of primary tumor locations were the oral cavity (49%), followed by the oropharynx (36%), hypopharynx (11%), and larynx (4%). P16 status was also evaluated in some of our patients with 9% positive and 24% negative. Nearly 70% of our patients were initially diagnosed to have stage III-IV disease. As for previous treatment, 60% of our patients had radical surgery and 73% of patients underwent chemoradiotherapy before the recurrent or metastatic disease. After recurrence or metastasis, 60% of patients had distant metastasis with or without local recurrence, while the remaining had locally recurrent disease only. Furthermore, 45% of patients had PD-L1 high expression while 55% of patients had PD-L1 low or no expression. Patients were then stratified according to the treatment sequence. There were 40 patients treated with Cet-IO and 35 patients treated with IO-Cet. PD-L1 expression levels were higher in IO-Cet than in Cet-IO insignificantly. Other basic characteristics include gender, age, primary tumor location, p16 status, initial stage, previous treatment history, and disease status upon enrollment well balanced between the two treatment arms.

### 3.2. Survival Outcomes

The median follow-up interval was 16 months. At the end of our study, 52% of our patients died and cancer was the main reason for their death. The oncologic outcomes were summarized in Table 2. The median PFS1 was 5.1 months in Cet-IO and 4.5 months in IO-Cet (*p* = 0.777) and the median PFS2 was 16.5 months in Cet-IO and 11.4 months in IO-Cet (*p* = 0.566). The median OS was 23.7 months versus 22.8 months in Cet-IO and IO-Cet, respectively (*p* = 0.484). The survival curves of PFS1, PFS2, and OS were plotted in Figure 1 and Figure 2. The ORR and DCR of first-line and second-line treatment were all higher in Cet-IO than in IO-Cet. In first-line treatment, the ORR and DCR were 73% versus 37% (*p* = 0.002) and 78% versus 63% (*p* = 0.165) in Cet-IO and IO-Cet, respectively. In second-line treatment, the ORR and DCR were 63% versus 37% (*p* = 0.028) and 78% versus 51% (*p* = 0.018) in Cet-IO and IO-Cet, respectively. Cox regression analyses with survival for potential prognostic factors were depicted in Table 3. Multivariate analysis demonstrated that primary tumor locations were independently negative predictors that correlated with survival. The treatment sequence, Cet-IO or IO-Cet, did not have a significant impact on survival. The relationship between PD-L1 expression and survival was also investigated. The survival curves of OS stratified by PD-L1 expression and treatment sequence were plotted in Figure 3. In patients with PD-L1 low expression, the median OS was not reached (NR) versus 22.8 months in Cet-IO and IO-Cet (*p* = 0.591), while in patients with PD-L1 high expression, the median OS was 23.7 months versus 29.4 months in Cet-IO and IO-Cet (*p* = 0.680).

## 4. Discussion

To our best knowledge, this is the first study investigating the oncologic outcomes of cetuximab followed by immunotherapy versus immunotherapy followed by cetuximab in patients with R/M HNSCC. Current guidelines only recommended that both cetuximab and immunotherapy were effective treatments in patients with R/M HNSCC. However, there was no clear evidence focusing on the treatment sequences. Little was known regarding the optimal treatment sequence. Thus, we conducted an observational study to compare Cet-IO with IO-Cet in patients with R/M HNSCC. In terms of PFS1, PFS2, and OS, our results suggested that the survival was insignificant between Cet-IO and IO-Cet. The median PFS1 was 5.1 months in Cet-IO and 4.5 months in IO-Cet (*p* = 0.777) and the median PFS2 was 16.5 months in Cet-IO and 11.4 months in IO-Cet (*p* = 0.566). The median OS was 23.7 months versus 22.8 months in Cet-IO and IO-Cet, respectively (*p* = 0.484). Nevertheless, the ORR of first-line treatment was much higher in Cet-IO than those in IO-Cet, accounting for 73% versus 37% (*p* = 0.002), respectively. Based on our results, both Cet-IO and IO-Cet were effective treatment sequences in patients with R/M HNSCC. Higher ORR in Cet-IO indicated that Cet-IO might be more suitable for patients who had large tumor burdens and urgent needs for treatment responses. Multivariate regression analysis also demonstrated that the treatment sequence, Cet-IO or IO-Cet, did not have a different significance on survival. Furthermore, PD-L1 expression was not a predictive biomarker in our study. Regardless of PD-L1 expression, Cet-IO exhibited insignificant survival in comparison with IO-Cet. In patients with PD-L1 low expression, the median OS was not reached (NR) versus 22.8 months in Cet-IO and IO-Cet, respectively (*p* = 0.591), while in patients with PD-L1 high expression, the median OS was 23.7 months versus 29.4 months in Cet-IO and IO-Cet, respectively (*p* = 0.591). Further prospective randomized control trials were warranted to validate our results.

Although both Cet-IO and IO-Cet were effective in patients with R/M HNSCC, there existed more evidence in Cet-IO. The extreme study was the pivotal phase 3 clinical trial demonstrating that the cetuximab-containing regimen was active first-line chemotherapy in patients with R/M HNSCC [5]. This study indicated that Cisplatin, 5-FU plus cetuximab followed by cetuximab maintenance weekly had significantly longer survival than cisplatin plus 5-FU. TPExtreme study also disclosed in a child that TPEx was non-inferior to Pembrolizumab with more tolerable treatment-related toxicities [11]. Median follow-up was 34.4 months in the TPEx group and 30.2 months in the EXTREME group. There were also several real-world studies investigating the efficacy of cetuximab as a first-line treatment in R/M HNSCC. Yoshino et al. evaluated the efficacy and toxicity of the first-line cetuximab-containing regimen in 33 Japanese patients and showed comparable results with PFS 4.1 months, OS 14.1 months, ORR 36%, and DCR 88% [12]. Mello et al. retrospectively recruited 121 European R/M HNSCC patients and suggested that a first-line cetuximab-containing regimen is a good treatment option with PFS 8 months, OS 11 months, ORR 24%, and DCR 49% [13]. Depenni et al. conducted an observational study with 297 R/M HNSCC and indicated that a first-line cetuximab-containing regimen is a new treatment modality with PFS 4.8 months and OS 10.8 months [14]. Tourneau et al. published their results of ENCORE which was a multinational, observational, prospective, open-label study focusing on a cetuximab-containing regimen in first-line 221 R/M SCCHN patients. Median PFS was 6.5 months and the median OS was 10.2 months [15]. Sano et al. examined the real-world treatment outcomes of the EXTREME regimen as first-line therapy for 100 R/M HNSCC patients and found the treatment response of the EXTREME regimen in Japanese patients was effective with PFS 5 months and OS 11 months [16]. These publications all confirmed the role of cetuximab as a first-line treatment in R/M HNSCC patients.

After a progression of first-line treatment, immunotherapy plays a crucial role in the second-line setting. Checkmate 141 demonstrated that nivolumaba as a second-line treatment resulted in significantly longer OS in patients with platinum-refractory R/M HNSCC [9], with median OS was 7.5 months versus 5.1 months in the nivolumab group and standard therapy group, respectively. Keynote-040 also confirmed the significant prolongation of OS of pembrolizumab in patients with platinum-refractory R/M HNSCC [10], with median OS was 8.4 months versus 6.9 months in pembrolizumab and standard of care, respectively. Furthermore, Lien et al. analyzed the subsequent treatment pattern of R/M HNSCC patients receiving a first-line cetuximab-containing regimen and found that second-line immunotherapy could improve OS, even in platinum-resistant populations [17]. Kariya et al. conducted a retrospective observational study in a real-world setting to investigate the effectiveness of nivolumab monotherapy followed by prior cetuximab treatment [18]. The median PFS was 2.0 months and the median OS was up to 8.0 months. More recently, a prospective phase I/II study to evaluate pembrolizumab plus docetaxel in platinum-refractory R/M HNSCC patients showed promising activity with a manageable safety profile [19]. The median PFS was 5.8 months and the median OS was 21.3 months. The ORR was 22.7% and DCR was 54.6%. This literature all supports the sequence of Cet-IO. Moreover, an interesting study evaluating the immune responses induced by cetuximab in tumor microenvironments also prefers cetuximab first followed by immunotherapy [20]. This study demonstrated more inflammatory cell aggregation in the chemotherapy with the cetuximab group, as well as more destruction of cancer cell foci and a higher infiltration of lymphocytes. More immune cell infiltration will make cold tumors into hot tumors. Thereafter, second-line immunotherapy might augment the treatment response in such a hot tumor.

As for IO-Cet, the main hypothesis is that HNSCC is considered as “immune deserts” which results in immune escape. Immunotherapies can stimulate the immune system, inhibit the immunosuppressive tumor microenvironment and enhance treatment responses of subsequent therapies [21]. Keynote-048 disclosed that pembrolizumab alone significantly improved OS versus cetuximab with chemotherapy in CPS of 1 or more population and pembrolizumab with chemotherapy also significantly improved OS versus cetuximab with chemotherapy in the total population [6]. With regard to subsequent treatment in a second-line setting, there was no solid evidence at present. Subgroup analysis of keynote-048 has shown an improved PFS2 (defined as the progression after the next line of therapy) in R/M HNSCC after a first-line pembrolizumab plus chemotherapy, suggesting a potential long-lasting effect of immunotherapy in the first-line setting [22]. Recently, Camarero et al. conducted a retrospective study and aimed to evaluate the efficacy of cetuximab after immunotherapy [23]. Overall, 23 R/M HNSCC patients were enrolled with a median PFS of 6 months and median OS of 12 months finally. Consistently, our study also concluded that IO-Cet was also an effective regimen for R/M HNSCC patients. Given the limited evidence at present, more clinical trials are needed to confirm the efficacy of the IO-Cet sequence in R/M HNSCC.

There were several potential limitations in this study that were inherent to any retrospective studies. First, our study only enrolled R/M HNSCC patients receiving both cetuximab and immunotherapy sequentially. Patients who did not receive subsequent second-line treatment after the progression of first-line cetuximab or immunotherapy were excluded. This might be a major bias in this study. Second, this study was a retrospective study with a nonrandomized design, and the treatment sequences of our patients were decided at the physician’s discretion, rather than randomly. This may influence the generalizability of our results. Third, our study enrolled some patients with p16+ oropharyngeal squamous cell carcinoma (OPSCC). Currently, increasing evidence suggested that p16+ OPSCC should be considered a different disease entity in HNSCC. Thus, treatment sequences in patients with p16+ OPSCC merited investigation separately in the future. Finally, a single institutional experience and a small sample size also limit the power of our study. Our study aimed to investigate the optimal treatment sequences in R/M HNSCC patients. To date, there was literature regarding the comparison of sequencing. Given that our retrospective study had several inevitable selection biases, our study provided the first real-world evidence to compare oncologic outcomes between Cet-IO and IO-Cet in R/M HNSCC patients. Further prospective randomized control studies were warranted to validate our conclusions.

## 5. Conclusions

Our study compared the oncologic outcomes between Cet-IO and IO-Cet in patients with R/M HNSCC. Based on our results, we disclosed that both Cet-IO and IO-Cet are effective in patients with R/M HNSCC with insignificant survival differences. The higher ORR of Cet-IO might render it to be considered in patients with large tumor burdens and urgent needs for treatment responses. In our multivariate analysis, primary tumor locations were independent predictors that correlated with survival. Our conclusions were clinically valuable and provided the first real-world evidence in the treatment of R/M HNSCC patients. Further prospective randomized controlled trials are merited to validate our conclusions.

## Figures and Tables

**Figure 1 cancers-14-02351-f001:**
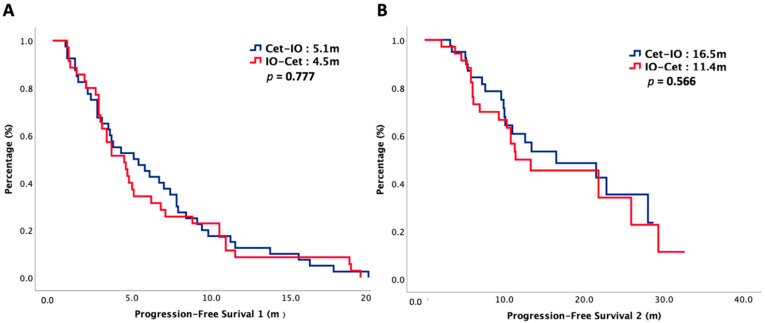
Progression-free survival 1 (**A**) and progression-free survival 2 (**B**) of 75 patients with R/M HNSCC, stratified by chemotherapy sequence.

**Figure 2 cancers-14-02351-f002:**
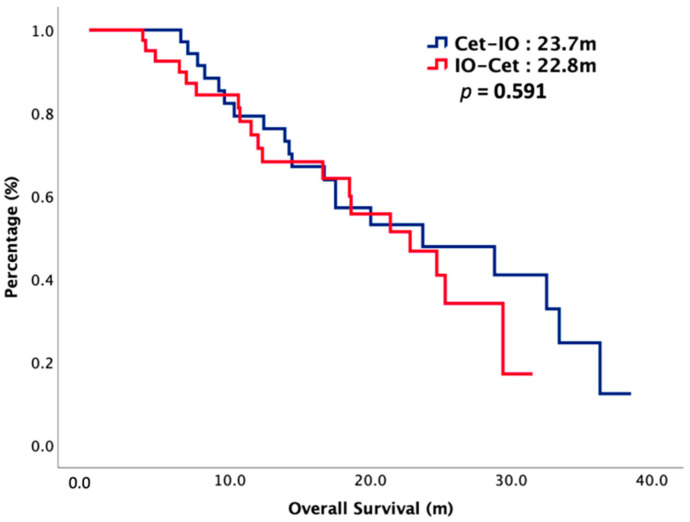
Overall survival of 75 patients with R/M HNSCC, stratified by chemotherapy sequence.

**Figure 3 cancers-14-02351-f003:**
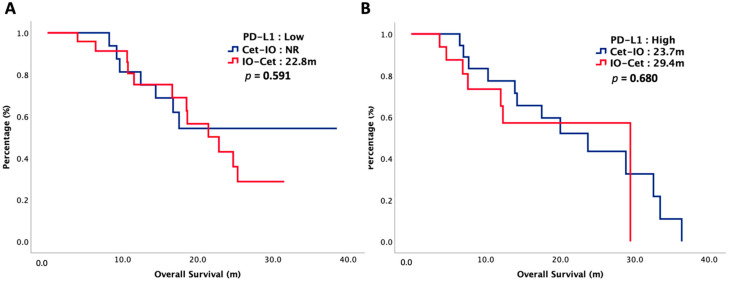
OS of 75 patients with R/M HNSCC, stratified by chemotherapy sequence and PD-L1 expression. (**A**): PD-L1: low; (**B**): PD-L1: High.

**Table 1 cancers-14-02351-t001:** Baseline clinical characteristics of 75 patients with R/M HNSCC, stratified by chemotherapy sequence.

	IO-Cet		Cet-IO		*p*
	*n* = 35		*n* = 40		
Gender					0.506
Male	34	97%	36	90%	
Female	1	3%	4	10%	
Age					0.948
≤60	23	66%	26	65%	
>60	12	34%	14	35%	
Primary tumor location					0.812
Oral cavity	19	54%	18	45%	
Oropharynx	11	31%	16	40%	
Hypopharynx	4	11%	4	10%	
Larynx	1	4%	2	5%	
p16 status					0.838
Positive	4	11%	3	8%	
Negative	8	23%	10	25%	
Unknown	23	66%	27	67%	
Initial T stage					0.279
T1–T2	14	40%	21	53%	
T3–T4	21	60%	19	47%	
Initial N stage					0.138
N0–N1	15	43%	24	60%	
N2–N3	20	57%	16	40%	
Initial M stage					0.402
M0	31	89%	39	97%	
M1	4	11%	1	3%	
Initial stage					0.170
I–II	8	23%	15	38%	
III–IV	27	77%	25	62%	
Previous surgery					1.000
Yes	21	60%	24	60%	
No	14	40%	16	40%	
Previous radiotherapy					0.189
Yes	28	80%	27	68%	
No	7	20%	13	32%	
Disease status upon enrollment					0.195
Local recurrence only	12	34%	18	45%	
Distant metastasis	23	66%	22	55%	
PD-L1 status					0.321
High expression	18	51%	16	40%	
Low expression	17	49%	24	60%	

R/M HNSCC, recurrent or metastatic head and neck squamous cell carcinoma; IO-Cet, immunotherapy-containing regimen followed by cetuximab-containing regimen; Cet-IO, Cetuximab-containing regimen followed by immunotherapy-containing regimen; PD-L1, programmed death ligand 1.

**Table 2 cancers-14-02351-t002:** Oncologic outcomes of 75 patients with R/M HNSCC, stratified by chemotherapy sequence.

	IO-Cet *n* = 35	Cet-IO *n* = 40	*p*
mPFS1 (m)	4.5	5.1	0.777
mPFS2 (m)	11.4	16.5	0.566
mOS (m)	22.8	23.7	0.484
First-line treatment			
CR (%)	2 (6)	8 (20)	
PR (%)	11 (31)	21 (53)	
SD (%)	9 (26)	2 (5)	
PD (%)	13 (37)	9 (22)	
ORR (%)	13 (37)	29 (73)	0.002
DCR (%)	22 (63)	31 (78)	0.165
Second-line treatment			
CR (%)	1 (3)	3 (8)	
PR (%)	12 (34)	22 (55)	
SD (%)	5 (14)	6 (15)	
PD (%)	17 (49)	9 (22)	
ORR (%)	13 (37)	25 (63)	0.028
DCR (%)	18 (51)	31 (78)	0.018

R/M HNSCC, recurrent metastatic head and neck squamous cell carcinoma; IO-Cet, immunotherapy-containing regimen followed by cetuximab-containing regimen; Cet-IO, Cetuximab-containing regimen followed by immunotherapy-containing regimen; mPFS1, median progression-free survival 1; mPFS2, median progression-free survival 2; mOS, median overall survival; CR, complete response; PR, partial response; SD, stable disease; PD, progressive disease; ORR, objective response rate; DCR, disease control rate.

**Table 3 cancers-14-02351-t003:** Cox regression analysis of parameters associated with survival.

Variables	HR (95% CI)	*p* Value
Gender, Male vs. Female	0.57 (0.22–1.47)	0.242
Age, ≤60 vs. >60	0.91 (0.46–2.82)	0.789
Primary tumor location, oral cavity vs. others	0.72 (0.36–0.80)	0.020
Initial T stage, T1–T2 vs. T3–T4	0.72 (0.36–1.41)	0.338
Initial N stage, N0–N1 vs. N2–N3	0.61 (0.31–1.19)	0.145
Initial M stage, M0 vs. M1	0.45 (0.14–1.49)	0.192
Initial stage, stage 1–2 vs. 3–4	0.99 (0.51–1.09)	0.969
p16 status, positive vs. negative	0.95 (0.44–2.03)	0.886
Previous radical surgery, yes vs. no	0.87 (0.45–1.67)	0.667
Previous radiotherapy, yes vs. no	0.84 (0.44–1.62)	0.606
Disease status, local only vs. metastasis	0.71 (0.31–1.36)	0.299
PD-L1 expression, high vs. low	0.76 (0.40–1.42)	0.383
Treatment sequence, Cet-IO vs. IO-Cet	0.79 (0.41–1.53)	0.485

HR, hazard ratio; CI, confidence interval; ECOG PS, Eastern Cooperative Oncology Group Performance Status; PD-L1, programmed death ligand 1; IO-Cet, immunotherapy-containing regimen followed by cetuximab-containing regimen; Cet-IO, Cetuximab-containing regimen followed by immunotherapy-containing regimen.

## Data Availability

The data presented in this study are available on request from the corresponding author. The data are not publicly available because are the propriety of E-Da Cancer Hospital.

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
