# Peer review of "Treatment Sequences in Patients with Recurrent or Metastatic Head and Neck Squamous Cell Carcinoma: Cetuximab Followed by Immunotherapy or Vice Versa"

_cancers, 2022, doi:10.3390/cancers14102351_

Round 1

Reviewer 1 Report

Yang et al. retrospectively analyzed the treatment outcome of head and neck squamous cell carcinoma patients treated sequentially with immune checkpoint inhibitors (ICIs; Nivolumab or Pembrolizumab) and the epidermal growth factor inhibitor cetuximab. Because it is arguably universal principle that cytotoxic/targeted regimens boost the effectiveness of ICIs, a reasonable next clinical question would be what sequence could be beneficial.

In order to address the clinical question, this single institution-based, retrospective study was performed. Although there are many limitations to drawing definitive conclusions, this study may add a piece of evidence for clinical decision-making.

Here are the comments.

1. L83–L86; the expression levels of PD-L1 substantially differ depending on the microenvironment. The information regarding the sites in which PD-L1 expression levels were determined should be provided, and if there are any criteria, the information should also be disclosed.

2.L113–L129: Overall response rates in the Cet-IO cohort excelled in the IO-Cet cohort, leading to the authors’ statement that Get-IO might be suitable for patients with a large tumor burden urgent needs for treatment responses. The reviewer thinks the positive finding could be added to the conclusions, as the observations may be a practical guide for clinical decision-making.

3. English proof editing. L82, L86: physicians’ discretion, rather than discrete of physicians; L65, L185, L206, L260, L268: pieces of evidence rather than evidences; L145: expression levels rather than expression; L83 expression levels rather than expression level.

Author Response

Reviewer #1

Thank you for your kind review. Here are our explanations. 

  1. L83–L86; the expression levels of PD-L1 substantially differ depending on the microenvironment. The information regarding the sites in which PD-L1 expression levels were determined should be provided, and if there are any criteria, the information should also be disclosed.

 Ans: Thanks for your kind suggestion. We had added more information in our manuscript as below. “econd-line immunotherapy was limited to pembrolizumab or nivolumab, which was determined by PD-L1 expression levels of primary tumors. Most tissues were resected specimens and the others were biopsies from patients with de novo metastatic HNSCC. PD-L1 scoring was calculated as the percentage of tumor cells exhibiting positive mem-brane staining at any intensity. Tumor proportional score (TPS) and combined positive score (CPS) PD-L1 expression were determined by Dako 22C3 pharmDx immunohisto-chemistry (IHC) assay, while tumor cell (TC) PD-L1 expression was determined by Dako 28-8 pharmDx IHC assay. High PD-L1 expression was defined as either TPS ≥50%, CPS ≥1 or TC ≥10%. Low PD-L1 expression was defined as TPS <50%, CPS <1 or TC <10%.”

2.L113–L129: Overall response rates in the Cet-IO cohort excelled in the IO-Cet cohort, leading to the authors’ statement that Get-IO might be suitable for patients with a large tumor burden urgent needs for treatment responses. The reviewer thinks the positive finding could be added to the conclusions, as the observations may be a practical guide for clinical decision-making.

Ans: Thank you for your kind suggestion. We had added this finding to our conclusion. “The robust ORR of Cet-IO renders Cet-IO more suitable for patients with large tumor burden and urgent needs for treatment responses.”

  1. English proof editing. L82, L86: physicians’ discretion, rather than discrete of physicians; L65, L185, L206, L260, L268: pieces of evidence rather than evidences; L145: expression levels rather than expression; L83 expression levels rather than expression level.

Ans: Thank you for your careful review. We had corrected them all.

Reviewer 2 Report

This retrospective, observational study investigated survival differences in patients with R/M SCCHN treated with a sequential chemotherapy-cetuximab and immune checkpoint inhibitor-chemotherapy regimens or vice versa. The report is well-written, concise and of significant clinical value for oncologists and patients. Currently, PD-L1 expression is the only predictive biomarker employed for selecting R/M SCCHN patients first-line treatment with pembrolizumab versus cetuximab and chemotherapy. The fact that, as cited by the authors "ORR of first-line treatment was much higher in Cet-IO than those in IO-Cet, accounting for 73% versus 37% (p = 0.002), respectively... Higher ORR in Cet-IO indicated that Cet-IO might be more suitable for patients who had large tumor burden and urgent needs for treatment responses.",  is highly valuabe information for the oncologist and their patients in treatment-decision making, which could render PD-L1 as the stand-alone predictive biomarker of less significance for patients with these bulky tumors. The authors are also well-aware about the flaws of their study as described in the discusson, but I still have some concerns regarding the methodology and selection-criteria for this population.

  • 83; The CPS is not reported as %, this has been stated in the 22C3 PharmDx Dako protocol for IHC PD-L1 scoring. Furthermore, why was the PD-L1 treshold for treatment eligibility with pembrolizumab set at 10? Current guidelines indicate a CPS 1 should be sufficient for treatment? How did you (statistically) determine this treshold? Furthermore, please elaborate on the used tissue specimen (primary tumors versus metastases? Biopsies versus resected specimens), the employed IHC test for PD-L1 testing and the staining platform.
  • 97; Why was cisplatin 5FU cetuximab administered according to a 4 week cycle and not a 3 week cycle as descibed in EXTREME? Furthermore, patients treated in the IO-Cet group received cetuximab + taxol in second-line while patients in the Cet-IO group received pembrolizumab with OR without taxol. Could you elaborate on the decision for not treating these patients with taxol? Toxicity? Low performance status? High PD-L1 expression?
  • 137 "p16 was evaluated in some patients". Please clarify. Patients with OPSCC? In addition, p16+ OPSCC are considered a different disease entity in SCCHN according to the 8th AJCC TNM staging due to their improved prognosis in comparison to p16- OPSCC and SCC at other anatomical sites. Ideally, the p16+ OPSCC should be investigated separately as a subcohort or it should be mentioned in the discussion section as an additional bias.
  • Although difficult to assess in a retrospective design, evaluating discrepancies on treatment-related toxicity in both Cet-IO and IO- Cet groups would also prove valuable for enhancing the quality of this study.

Some minor additional comments:

  • The introduction is concise but some terms should be explained a bit more (e.g. cetuximab, an EGFR-inhibitor; pembrolizumab, an anti-PD-1 agent...)
  • 36. "The median overall survival (OS) was around 8-10
    months in the past": please elaborate a bit more on this. Until what year? Over which period? Do you mean median survival across all ages and anatomical sites? You may also mention that during 20-30 years, the therapeutic landscape in SCCCHN remained unchanged resulting in similar survival rates, until the introduction of the EXTREME regimen...
  • 43-44, please replace by 'Other novel agents in SCCHN were immune checkpoint inhibitors such as the anti-PD-1 agent, pembrolizumab: this was introduced as a first line treatment in R/M SCCHN..."
  • please provide a simple summary as instructed by the journal's author instructions.
  • please check for spelling errors throughout the manuscript

Author Response

Reviewer #2

Thank you for your kind review. Here are our explanations. 

  • 83; The CPS is not reported as %, this has been stated in the 22C3 PharmDx Dako protocol for IHC PD-L1 scoring. Furthermore, why was the PD-L1 treshold for treatment eligibility with pembrolizumab set at 10? Current guidelines indicate a CPS 1 should be sufficient for treatment? How did you (statistically) determine this treshold? Furthermore, please elaborate on the used tissue specimen (primary tumors versus metastases? Biopsies versus resected specimens), the employed IHC test for PD-L1 testing and the staining platform.

Ans: Thank you for your careful review. We apologized for our mistyping. The accurate threshold of CPS in our study was 1 , not 10%. We had corrected it. Most specimens were resected primary tumors. The others were biopsies from patients with de novo metastatic HNSCC. Tumor proportional score (TPS) and combined positive score (CPS) PD-L1 expression were determined by Dako 22C3 pharmDx immunohistochemistry (IHC) assay, while tumor cell (TC) PD-L1 expression was determined by Dako 28-8 pharmDx IHC assay. We had added all information to our manuscript.

  • 97; Why was cisplatin 5FU cetuximab administered according to a 4 week cycle and not a 3 week cycle as descibed in EXTREME? Furthermore, patients treated in the IO-Cet group received cetuximab + taxol in second-line while patients in the Cet-IO group received pembrolizumab with OR without taxol. Could you elaborate on the decision for not treating these patients with taxol? Toxicity? Low performance status? High PD-L1 expression?

Ans: Thank you for your suggestion. The EXTREME was commonly administrated in a 4- week cycle in Asia patients because of toxicity. Given that there was no clear evidence regarding to combination of immunotherapy and taxane in R/M HNSCC, adding taxane to immunotherapy in second-line treatment was determined at physicians’ discretion.  

  • 137 "p16 was evaluated in some patients". Please clarify. Patients with OPSCC? In addition, p16+ OPSCC are considered a different disease entity in SCCHN according to the 8th AJCC TNM staging due to their improved prognosis in comparison to p16- OPSCC and SCC at other anatomical sites. Ideally, the p16+ OPSCC should be investigated separately as a subcohort or it should be mentioned in the discussion section as an additional bias

Ans: Thank you for your suggestion. P16 was evaluated routinely in OPSCC in our study, not all HNSCC. So, we just presented the P16 data of some patients. Meanwhile, we agree with your opinion that p16+ OPSCC was different from other HNSCC and may need investigation separately. Thus, we had added this biasis in our discussion as below. “Third, our study enrolled some patients with p16+ oropharyngeal squamous cell carcinoma (OPSCC). Currently, increasing evidence suggested p16+ OPSCC should be considered a different disease entity in HNSCC. Thus, treatment sequences in patients with p16+ OPSCC merited investigation separately in the future.”

  • Although difficult to assess in a retrospective design, evaluating discrepancies on treatment-related toxicity in both Cet-IO and IO- Cet groups would also prove valuable for enhancing the quality of this study.

Ans: Thank you for your suggestion. Given that this was a retrospective study, it was difficult to realize the treatment-related toxicity. Actually, a prospective study is ongoing to answer this question.

Some minor additional comments:

  • The introduction is concise but some terms should be explained a bit more (e.g. cetuximab, an EGFR-inhibitor; pembrolizumab, an anti-PD-1 agent...)

Ans: Thank you for your suggestion. We had add more information in our introduction as your suggestion.

  • "The median overall survival (OS) was around 8-10
    months in the past": please elaborate a bit more on this. Until what year? Over which period? Do you mean median survival across all ages and anatomical sites? You may also mention that during 20-30 years, the therapeutic landscape in SCCCHN remained unchanged resulting in similar survival rates, until the introduction of the EXTREME regimen...

Ans: Thank you for your suggestion. The median OS of R/M HNSCC was about 8-10 months before 2005.

  • 43-44, please replace by 'Other novel agents in SCCHN were immune checkpoint inhibitors such as the anti-PD-1 agent, pembrolizumab: this was introduced as a first line treatment in R/M SCCHN..."

Ans: Thank you for your suggestion. We had revised our sentences as your suggestion.

  • please provide a simple summary as instructed by the journal's author instructions.

Ans: A simple summary was provided as instructions.

  • please check for spelling errors throughout the manuscript

Ans: We had carefully checked our spelling errors as possible. Thank you.

Round 2

Reviewer 2 Report

The authors have responded adequately to my questions and sufficiently adapted the manuscript. The paper can be published.

Author Response

Thank you for your kind review!!